# Does Adding Standard Systematic Biopsy to Targeted Prostate Biopsy in PI-RADS 3 to 5 Lesions Enhance the Detection of Clinically Significant Prostate Cancer? Should All Patients with PI-RADS 3 Undergo Targeted Biopsy?

**DOI:** 10.3390/diagnostics11081335

**Published:** 2021-07-26

**Authors:** Enrique Gomez-Gomez, Sara Moreno Sorribas, Jose Valero-Rosa, Ana Blanca, Juan Mesa, Joseba Salguero, Julia Carrasco-Valiente, Daniel López-Ruiz, Francisco José Anglada-Curado

**Affiliations:** 1Maimonides Institute of Biomedical Research of Cordoba (IMIBIC), 14004 Cordoba, Spain; sara.moreno92@gmail.com (S.M.S.); jose.valero.rosa.sspa@juntadeandalucia.es (J.V.-R.); anblape78@hotmail.com (A.B.); josebasalguerosegura@gmail.com (J.S.); julia.carrasco.sspa@juntadeandalucia.es (J.C.-V.); danielj.lopez.sspa@juntadeandalucia.es (D.L.-R.); franciscoj.anglada.sspa@juntadeandalucia.es (F.J.A.-C.); 2Department of Urology, Reina Sofia University Hospital (HURS), 14004 Cordoba, Spain; 3Radiology Department, Reina Sofia University Hospital (HURS), 14004 Cordoba, Spain; juanmeque@gmail.com

**Keywords:** prostate imaging reporting and data system (PI-RADS), MRI targeted biopsy, target-ing plus standard biopsy, PI-RADS 3 lesions

## Abstract

Introduction. Our aim was to assess the value of adding standard biopsy to targeted biopsy in cases of suspicious multiparametric magnetic resonance imaging (mp-MRI) and also to evaluate when a biopsy of a PI-RADS 3 lesion could be avoided. Methods: A retrospective study of patients who underwent targeted biopsy plus standard systematic biopsy between 2016–2019 was performed. All the 1.5 T magnetic resonance images were evaluated according to PI-RADSv.2. An analysis focusing on the clinical scenario, lesion location, and PI-RADS score was performed. Results. A total of 483 biopsies were evaluated. The mean age was 65 years, with a PSA density of 0.12 ng/mL/cc. One-hundred and two mp-MRIs were categorized as PI-RADS-3. Standard biopsy was most helpful in detecting clinically significant prostate cancer (csPCa) in patients in the active surveillance (AS) cohort (increasing the detection rate 12.2%), and in peripheral lesions (6.5%). Adding standard biopsy showed no increase in the detection rate for csPCa in patients with PI-RADS-5 lesions. Considering targeted biopsy in patients with PI-RADS 3 lesions, a higher detection rate was shown in biopsy-naïve patients versus AS and in patients with a previous negative biopsy (*p* = 0.002). Furthermore, in these patients, the highest rate of csPCa detection was in anterior lesions [42.9% (*p* = 0.067)]. Conclusions. Our results suggest that standard biopsy could be safely omitted in patients with anterior lesions and in those with PI-RADS-5 lesions. Targeted biopsy for PI-RADS-3 lesions would be less effective in peripheral lesions with a previous negative biopsy.

## 1. Introduction

Prostate cancer is the most common cancer in developed countries [1].

The diagnosis of prostate cancer has evolved in the last several years, due to the low accuracy of the classic PSA pathway: digital rectal examination (DRE) and standard biopsy [2].

First, different tools, such as risk calculators that combine different clinical variables, have been shown to outperform the classic pathway [3,4].

Furthermore, different non-invasive biomarkers have been developed to improve the detection of clinically significant prostate cancer (csPCa), defined mostly as a Gleason score ≥ 7 [5].

Finally, clear and high-quality evidence has emerged, which shows that the introduction of multiparametric magnetic resonance (mpMR) further improves the classic pathway, with a higher detection rate of csPCa when targeting lesions described as positive with mpMR (PI-RADS or LIKERT ≥ 3) [6,7,8].

mp-MRI has not only improved the detection of csPCa but also has helped to avoid unnecessary biopsies because of its high negative predictive value [9]. However, beyond the known limitations, i.e., the learning curve and the need for homogeneous training criteria, any lesion defined as equivocal (PI-RADS 3) is also considered a limitation and because a low percentage of PI-RADS 3 cases are found to harbor a csPCa. This results in a very low positive predictive value (PPV) [10]. Thus, it seems necessary to evaluate additional techniques for the adequate management of this group of patients [11]. One of the most studied of these techniques is the PSA density (PSAD), which, when combined with PI-RADS 3 lesions showed an improvement in the selection of patients for prostate biopsy [11]. However, heterogeneous evidence about specific lesion location and clinical scenarios have been reported [12,13,14,15]. In line with these issues, there appears to be value in adding standard biopsies to targeted biopsies. Current studies have shown that, in general, adding standard biopsies increased the detection rate of csPCa in around 5–10% of patients, at the cost of an over-diagnosis of insignificant tumours, along with the potential for biopsy-related complications as well as discomfort for the patient [16]. Depending on the patient’s clinical picture, incrementally increased detection rate that results from adding standard biopsies or the over-diagnosis of insignificant tumours may vary and more data are needed to be able to clarify the best recommendations in each setting.

Given the above, PI-RADS 3 cases present a diagnostic and prognostic challenge. The main objective of our study was to assess the value of adding standard biopsy to targeted biopsy and to evaluate when a lesion biopsyies could be avoided based on lesion location and clinical scenarios when an mp-MRI discloses PI-RADS 3 (i.e., an equivocal or indeterminate lesion).

## 2. Materials and Methods

### 2.1. Population

A retrospective study in a prospectively collected cohort of biopsied patients in whom there was at least one suspicious mp-MRI lesion, from June 2016–December 2019, was performed. The study was approved by the local research ethics committee, and written, informed consent was obtained from all participants.

Inclusion criteria were:Indication of prostate biopsy because of a positive mp-MRI (PI-RADS ≥ 3).Patients had undergone a targeted + a standard biopsy.A pathology report based on the newly proposed grading prognostic group (ISUP grade group GGG)) was available [17].All the information about the variables selected for the study was available.All patients with an mp-MRI scan of poor quality, red, or those who were unavailable for review by an experienced uroradiologist were excluded.

### 2.2. Study Design and Objectives

A retrospective analytical study (based on a prospective database) was carried out. CsPCa was defined as any GGG ≥ 2, or a high PCa volume (three positive cores in the standard biopsy or ≥5 mm of maximum positive core length).

For this study, there were two main objectives:To analyze the diagnostic value of additional standard biopsy in the setting of targeted biopsy, in which the main endpoint was to evaluate the increase in the diagnoses of csPCa stratified according to the clinical setting and location of the lesion.To analyze the effectiveness of targeting PI-RADS 3 lesions, in which the main endpoint was to evaluate the percentage of csPCa detected at different lesion locations and clinical scenarios.

### 2.3. Mp-MRI Protocol and Characteristics

Prostate mp-MRI examinations were performed on a 1.5 T MRI system (Magne-tomAera, Siemens AG, Erlangen, Germany) using a 16-channel phased-array body coil (without an endorectal coil).

To minimize the number of confusing lesions caused by haemorrhages, the MRI scan was performed at least eight weeks after any previous prostate biopsy. The prostate mp-MRI protocol included T1-weighted images (T1WI), T2-weighted images (T2WI) in three planes, diffusion-weighted images (DWI), and dynamic contrast-enhanced images (DCE). For further description of the protocol, see the Appendix A. Radiological reports were based on PI-RADS v.2; lesions were described and localized according to the division of the prostate sectors recommended by PI-RADS v.2. In the peripheral zone, lesions were measured on DWI with a calculation of the apparent diffusion coefficient (ADC), but, in the transition zone, lesions were measured using T2WI. Finally, a PI-RADS Assessment Category was described for prostate findings. Prostate mp-MRI examinations were evaluated using a PACS (Vue PCAS, Carestream, 12.1.5.5151 version) and specific software (Syngo.via, MR prostate, Siemen AG, Erlangen, Germany). It should be mentioned that five radiologists initially analysed the mp-MRI results in clinical practice (including the experienced radiologist), but the most experienced radiologist (more than 500 MRIs and five years of experience) reviewed all images at a meeting before the biopsy. The reviewed version (PI-RADS score) was ultimately used to proceed. Only the index lesion was considered for the analysis.

### 2.4. Biopsy Technique

The biopsy technique included an MRI-ultrasound fusion biopsy using a General Electric ultrasound machine (LOGIQ E9, GE Healthcare: Milwaukee, Wisconsin EE.UU) with real-time sensor-based software, as previously described [18], as well as a standard 12-core biopsy in case of biopsy-naïve or previous negative biopsy, and an 18-core biopsy in the context of patients under active surveillance. Concerning the targeted biopsy, a minimum of three cores were obtained. Two uro-pathologists, who specialized in prostate cancer evaluation, reported biopsy findings using the newly proposed GGG. Briefly, three experienced urologists performed all biopsies using a transrectal approach. Targeted biopsy was always performed first by the same urologist who performed 12-core or 18-core sextant biopsy posteriorly. Further details of the biopsy technique are described in the Appendix A.

### 2.5. Variables and Statistical Analysis

A descriptive study was performed by calculating the median and interquartile ranges (IR) for the quantitative variables and the absolute frequencies and percentages for the qualitative variables.

### 2.6. Main Variables

Independent variables were: age, PSA, DRE, prostate volume, PSA density (PSAD) using prostate volume based on MRI, lesion location (peripheral, transition/central, anterior fibromuscular stroma), initial PI-RADS score from clinical practice (I-PI-RADSv.2 score), PI-RADS score after expert review (PI-RADSv.2 score), lesion diameter, clinical scenario (first biopsy, prior negative biopsyies, or active surveillance) and targeted biopsy cores sampled. For comparison between quantitative variables, a *t*-test for paired and unpaired data was used. Qualitative variables were evaluated with the chi-square or McNemar test. Multivariate logistic regression analysis was performed for adjusted analyses. Agreement among radiologists was evaluated by the kappa test. SPSS v.17 software was used to perform the analysis. A *p*-value of less than 0.05 was defined as showing for statistically significant differences.

## 3. Results

### 3.1. Clinical Characteristics of the Cohort

There were 483 biopsy procedures evaluated. The mean age of patients at the time of biopsy was 65 years. More than 75% of patients had a PSA level < 10 ng/mL. The most frequent PI-RADS score was PI-RADS 4. In addition, 14.9% of targeted lesions on mp-MRI were re-classified with a different PI-RADS score by the expert radiologist, revealing a kappa agreement of 0.76; *p* < 0.001.

There were 252 patients (52.7%) diagnosed with csPCa, with most because of a GGG ≥ 2 (213; 84.2%), 29 (11.5%) because of a maximum core length, and the rest (4.3%) because of the number of positive cores at standard biopsy. Further clinical details are shown in Table 1.

Targeted biopsy diagnosed more csPCa than the standard approach and maximum tumour length was also higher in targeted versus standard biopsy (47.4% vs. 34.8%; *p* < 0.001 and mean length 8.4 vs. 5.5 mm; *p* < 0.001). Targeted biopsy results according to the PI-RADS score are shown in Appendix A.

### 3.2. Value of Additional Standard Biopsy to Targeted Biopsy in PI-RADS 3–5

The diagnosis of csPCa increased statistically significantly from 47.4% to 52.2% when standard biopsy was added to targeted biopsy (*p <* 0.001). This change of approximately 5% was due to an upgrade in pathology from GGG 1 to GGG 2 (56.5%), and to an increase in the number of positive cores. However, with this statistically significant increase in csPCa, a similar 5% increase in non-significant PCa also occurred.

The clinical context associated with the highest detection of csPCA was in patients under active surveillance, patients with a prior negative biopsy, and patients undergoing their first biopsy, with detection rates of 12.2%, 8.8% and 3.2%, respectively.

Based on zone location, the highest increase in detection rate was 6.5% for peripheral zone lesions, 3% for the transition zone lesions and no increase for anterior fibromuscular stroma lesions. Figure 1 shows the increase in the detection rate was dependent on both the clinical scenario and the lesion location.

When analyzing according to the PI-RADS score, csPCa was detected only by systematic biopsy in 5/19 (26.3%), 14/148 (9.5%), and 0/79 (0%) for PI-RADS 3, 4, and 5 lesions, respectively. If only the GGG and tumour length were used to define csPCa, the detection rates when adding systematic biopsy to targeted biopsy were 4/13 (23.5%), 11/131 (7.7%), and 0/79 (0%) for PI-RADS 3, 4, and 5 lesions, respectively.

### 3.3. Value of Targeting PI-RADS 3 Lesions

In 102 patients, the mp-MRI was categorized as PI-RADS 3. The median PSAD was 0.10 (0.07–0.16), with the majority of the lesions in the peripheral zone. Further clinical details of this specific cohort are shown in Table 2.

In this analysis, the effectiveness of targeting PI-RADS 3 lesions clearly differed depending on the clinical scenario, with detection rates of csPCa of 44.4%, 6.8%, and 25% at the first biopsy, prior negative biopsy, and in the active surveillance cohort, respectively (*p* = 0.002). The differences between the lesion locations did not reach statistical significance, but a clear trend (*p* = 0.067), with detection rates of 11.4% vs. 12.5% vs. 42.9% in the peripheral, transition/central zone, and anterior fibromuscular stroma zones, respectively. PSAD (>0.15 ng/mL/cc) was not associated with a statistically significant higher detection rate (12.2 vs. 17.9%; *p* = 0.52).

Multivariate analysis showed a significant decrease in the detection rate of csPCa for targeted biopsy in patients having PI-RADS 3 lesions when zone location was the peripheral or transition/central zone vs. the anterior fibromuscular stroma zone [OR = 0.082 (0.008–0.885) and 0.036 (0.001–0.936), respectively]. A significantly decreased risk for csPCa also occurred in the clinical context of a prior negative biopsy vs. biopsy-naïve patients (OR = 0.033 (0.004–0.262) (Table 3)).

## 4. Discussion

Despite the reports on the value mp-MRI for prostate cancer diagnosis and biopsy guidance, there remain areas of debate, how best do we deal with PI-RADS 3 lesions, and when should standard biopsies be added to targeted biopsies.

In this research, both questions were evaluated through a cohort of more than 450 targeted biopsies. The clinical characteristics of our patients are in accord with most studies insofar of age, PSA, gland volume, and rate of PCa diagnosis. The target lesion detection rate is also in agreement with other published reports [19,20].

Our prevalence of PI-RADS 3 cases reported was around 20%, which is in line with that previously reported (17.3% [range 6.4–45.7%]) [20]. The detection of PCa and csPCa using targeted + standard biopsy was also similar globally in our PI-RADS 3 population. We had rates of 18.6% for csPCa and 36.3% for any PCa compared to literature reports of 18.5% for csPCa and 36% for any PCa [20,21,22,23]. However, our results do not confirm the results published by others concerning the value of a specific cut-off threshold of 0.15 for PSAD [21,22,24]. Using this threshold, we would have missed more than 10% of cases of csPCa. Instead, our results indicate that a patient’s prior biopsy status and lesion location should be considered in these equivocal PI-RADS 3 lesions. We provide data that indicates a lower probability of csPCa in patients with a prior negative biopsy vs. biopsy-naïve patients [11]. Our results also highlighted the relevance of lesion location, revealing that fibrostromal anterior lesions, which have not been sampled in patients with previous negative biopsies, have a higher probability of csPCa vs. lesions biopsied from the peripheral and transition/central zones.

When evaluating the role of adding a standard biopsy procedure to already obtained targeted biopsy, our results corroborate that such an additional biopsy results in a small increase in the diagnosis of csPCa, at the cost of an additional procedure, with associated risk to the patient and medical expenses, and more insignificant PCa detected [16].

What our results primarily suggest is that the best scenario in which to add the standard biopsy is likely in patients under active surveillance, as this is where a greater increase in csPCa detection can be achieved, with no additional cost of diagnosing non-csPCa (as this will have already been diagnosed) [25,26]. A recent review and meta-analysis by Baccaglini et al. showed similar positive predictive values for MRI-targeted biopsies alone vs. a combination of both targeted and standard biopsies in the context of active surveillance for patients with low-risk PCa. The authors concluded that the use of MRI-targeted biopsies alone may be preferable in this group of patients [27]. From our data, there are two clear situations in which there is no role for the addition of the standard biopsy approach—PI-RADS 5 lesions and in those lesions with an anterior fibromuscular stroma location. In line with this, Arabi et al. [28] showed that standard biopsy increased the detection rate in those patients with PI-RADS 3 and 4 lesions, but not in those with PI-RADS 5 lesions. In terms of lesion location, recent data in cohorts that used a targeted plus a standard biopsy approach, as reported by Falagario et al. [29], have shown that the probability of adding standard biopsy to targeted biopsy to detect csPCa in non-peripheral zone lesions is clearly inferior. One of the reasons for the additional csPCa detected by standard biopsy, specifically in the peripheral zone, and in smaller lesions, is not only mp-MRI missed diagnosis, but also targeting errors, as has been previously demonstrated [30]. This is why a saturation targeted approach has recently been highly recommended, specifically for smaller lesions [31].

From our data, we can suggest that, in a clinical practice setting, standard biopsies could be safely omitted in patients with anterior lesions and in those with PI-RADS 5 lesions. Targeted biopsy for PI-RADS 3 lesions could also be omitted in patients with peripheral lesions and in those with a previous negative biopsy. However, this data must be interpreted carefully, as some limitations should be considered: this is a retrospective analysis in a prospectively collected database from clinical practice; the biopsy-naïve patient cohort is limited; and finally, our standard biopsy approach is a transrectal 12-core biopsy, in which accuracy could be limited compared to the transperineal approach. Furthermore, our urologists were not blinded to mp-MRI information when standard biopsies were performed which could have affected the actual rate of detection of the standard biopsy approach. However, this reflects the real clinical practice, and most urologists follow the same standard biopsy protocol.

## 5. Conclusions

From a clinical practice perspective, with a targeted-biopsy cohort, we can suggest that standard biopsy could be safely omitted in patients with anterior lesions and in those with PI-RADS 5 lesions. Targeted biopsy for PI-RADS 3 lesions would be less effective in detecting csPCa and thus omitted in patients with peripheral zone lesions and a previous negative biopsy.

## Figures and Tables

**Figure 1 diagnostics-11-01335-f001:**
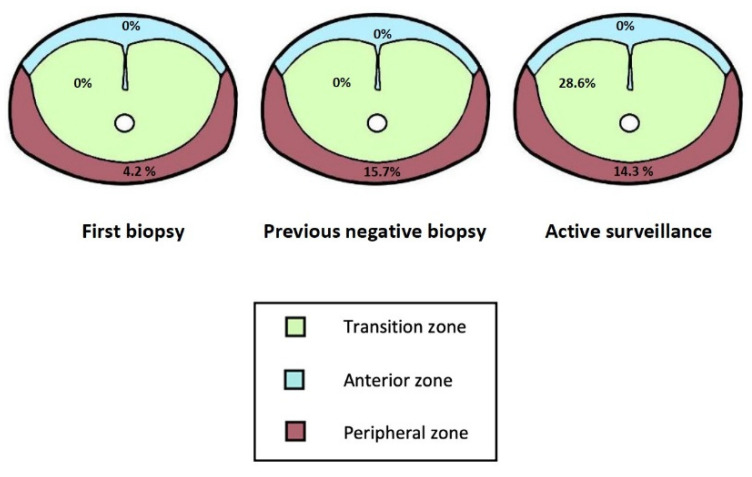
The additional detection rate for standard biopsy depending on both the clinical scenario and lesion location.

**Table 1 diagnostics-11-01335-t001:** Characteristics and description of the cohort according to csPCa status.

Variable	Total	csPCa	No csPCa
	*n* = 483	*n* = 252	*n* = 231
**Age; years**	65 (59–71)	68 (63–73)	63 (58–68)
**PSA; ng/mL**	6.39 (4.8–9.5)	6.7 (511–10.5)	5.85 (4.4–8.7)
**Prostate volume; cc**	55 (40–76)	50 (36–69)	60 (44–80.1)
**PSAD**	0.12 (0.08–0.18)	0.14 (0.10–0.23)	0.10 (0.07–0.15)
**Clinical scenario**			
1° Biopsy	45 (9.3)	31 (12.3)	14 (6.1)
Previous negative biopsy	337 (69.8)	159 (63.1)	178 (77.1)
Active surveillance	101 (20.9)	62 (24.6)	39 (16.9)
**mp-MRI I-PI-RADSv.2 score**			
3	105 (21.7)	21 (8.3)	84 (36.4)
4	289 (59.8)	153 (60.7)	136 (58.9)
5	89 (18.4)	78 (31)	11 (4.8)
**mp-MRI PI-RADSv.2 score**			
2	50 (10.4)	6 (2.4)	44 (19)
3	102 (21.1)	19 (7.5)	83 (35.9)
4	245 (50.7)	148 (58.7)	97 (42)
5	86 (17.8)	79 (31.3)	7 (3)
**Targeted lesion mm**	10 (7–14)	14 (10–22)	10 (7–13)
**Targeted location**			
Peripheral	325 (67.3)	155 (61.5)	170 (73.6)
Transition/central zone	66 (13.7)	26 (10.3)	40 (17.3)
Anterior fibromuscular stroma	92 (19)	71 (28.2)	21 (9.1)
**Number of targeted cores**	4 (4–5)	4 (3–4)	4 (4–5)

**Table 2 diagnostics-11-01335-t002:** Characteristics and description of the PI-RADSv.2 score 3 cohort.

Variable	Total
	*n* = 102
**Age; years**	62 (57–68)
**PSA; ng/mL**	5.8 (4.3–8)
**Prostate volume; cc**	57 (41–81)
**PSAD**	0.10 (0.07–0.16)
**Clinical scenario**	
1° Biopsy	9 (8.8)
Previous negative biopsy	73 (71.6)
Active surveillance	20 (19.6)
**Targeted lesion mm**	8 (6–10)
**Targeted location**	
Peripheral	79 (77.5)
Transition/central zone	16 (15.7)
Anterior fibromuscular stroma	7 (6.9)
**Number of targeted cores**	4(3–4)

Median and interquartile range for quantitative and absolute number and percentage for qualitative variables.

**Table 3 diagnostics-11-01335-t003:** Multivariate analysis showing the variables associated with a csPCa on the target biopsy in only the PI-RADS score 3 cohort.

VARIABLE	Multivariate Analysis for csPCa, *n* = 102		
	OR	*p*	95% CI (OR)
PSAD	1.389	0.692	0.272–7.085
Transition/central zone vs. anterior fibromuscular stroma	0.036	0.046	0.001–0.936
Peripheral zone vs. anterior fibromuscular stroma	0.082	0.039	0.008–0.885
Previous negative biopsy vs. biopsy-naïve setting	0.033	0.001	0.004–0.262
Active surveillance vs. biopsy-naïve setting	0.342	0.266	0.052–2.266

## Data Availability

The data presented in this study are available on request from the corresponding author.

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
