# Peer review of "Does Adding Standard Systematic Biopsy to Targeted Prostate Biopsy in PI-RADS 3 to 5 Lesions Enhance the Detection of Clinically Significant Prostate Cancer? Should All Patients with PI-RADS 3 Undergo Targeted Biopsy?"

_diagnostics, 2021, doi:10.3390/diagnostics11081335_

Round 1
Reviewer 1 Report
I have not done a superficial review of this article. Many hours were spent reading and re-reading the manuscript and retrieving some cited references as well as important articles on the topic of targeted & systematic biopsies. I have worked closely over more than a decade with one of the pioneers in mp-MRI (Barentsz). With this said, the submitted manuscript is difficult to read, and more difficult to fully comprehend. It does need major improvement in the structure of the article to allow the reader to grasp the THLs (Take Home Lessons). As it is currently written, this is not the case. I think the authors would greatly benefit if they prepared their manuscript using Word and had others with a better command of English show their recommendations using the "reviewer" tool in Word.
I would agree to offer assistance in the way the content is presented, but it is hard to do that by marking up a PDF.

Author Response
Response to the Editor and point-by-point responses to the Reviewer’s comments
We sincerely thank the Editor and Reviewers for their constructive comments and suggestions, which we consider to be very helpful in improving the quality of our manuscript. Accordingly, we have made many specific changes in the manuscript, based on these comments, as is explained in detail below in a point-by-point description of the changes introduced. Changes in the revised manuscript are highlighted in red.
We honestly trust that our responses and the modifications provided will help to strengthen the support of the Editor and reviewers, and hope that this revised version of our manuscript fits within the high-quality scientific standards of Diagnostics and, may, therefore, become acceptable for publication in the journal.
Reviewer 1
I have not done a superficial review of this article. Many hours were spent reading and re-reading the manuscript and retrieving some cited references as well as important articles on the topic of targeted & systematic biopsies. I have worked closely over more than a decade with one of the pioneers in mp-MRI (Barentsz). With this said, the submitted manuscript is difficult to read, and more difficult to fully comprehend. It does need major improvement in the structure of the article to allow the reader to grasp the THLs (Take Home Lessons). As it is currently written, this is not the case. I think the authors would greatly benefit if they prepared their manuscript using Word and had others with a better command of English show their recommendations using the "reviewer" tool in Word.
I would agree to offer assistance in the way the content is presented, but it is hard to do that by marking up a PDF.
Author’s response: First, we thank the reviewer for the time dedicated to improving the paper. We understand the reviewer’s concerns, and consequently, we have considered all the reviewer’s comments and the recommendations, and subsequently, we made all the changes in the manuscript and added the references suggested. Furthermore, the paper has been reviewed by Ms. Mary McAllister (scientific editor for 29 years for the Johns Hopkins University).
In the following lines, we explain and respond the reviewer’s comments and questions
Reviewer 1 comment: Most of the literature on MRI and its value in enhanced detection of significant prostate cancer is in the equivocal PI-RADS stage called PI-RADS 3. Therefore, the authors should try to be clearer as to their goal. Is it to see retrospectively if adding targeted biopsy to systematic (standard) biopsy is valuable (and to indicate the pros and cons based on their findings)?
Author’s response: We sincerely apologize for this misunderstanding and thank you for pointing this out. As currently the evidence favors targeted biopsy over standard biopsy, our main goal was to evaluate the role of adding this standard biopsy when a targeted biopsy is performed. Following the reviewer’s recommendation, the objective has been rewritten to make it clearer. Lines 57-59.
Reviewer 1 comment: The authors should use the same term and not switch to another name. How are they defining "standard biopsy?"
Since a major issue in this paper relates to significant PCa, where is its definition related to the reader? Lines 131-2 mention "higher" tumor length. The implication that this is a reason to call these lesions significant. If so, define the characteristics of significant in the paper.
Author’s response: We thank the reviewer for pointing this out. We have revised and clarified the definition in the Materials and Methods section. Lines 77-79 and 109-112.
Reviewer 1 comment: So the authors are equating "standard" biopsy with systematic biopsies. Yes? In the world of prostate cancer, the standard biopsy involved taking samples from each sextant of the prostate gland (i.e., right and left base, mid-gland and apex). If this is the case, somewhere in the Methods section this should be made clear or clearer (i.e., perhaps one section that is labeled "Definition of Terms."
Author’s response: Again, we sincerely apologize for this misunderstanding. Further description and definition have been added in the Materials and Methods section. Lines 109-112.
Reviewer 1 comment: A PSAD of 0.15 is considered the threshold for normal. What happened when higher PSAD's were looked at such as PSAD of 0.2, 0.25 or higher? Perhaps not enough cases. What about apparent diffusion coefficient (ADC) vs SigPCa? One of the most important parameters in multi-parametric MRI (mp-MRI) is the DWI (diffusion weighted imaging) and the reading of the apparent diffusion coefficient (ADC). The ADC is inversely correlated with the GGG, so the ADC should be a huge determinant of significant prostate cancer. The lower the ADC, the more aggressive the histology of the prostate biopsies.
Author’s response: Thank you for the interesting questions. Indeed, the PSA density value is an additional variable that must be considered, both in the clinical assessment and during the evaluation of the mpMRI of the prostate. As the reviewer correctly surmised, the median value in our sample was 0.10 (interquartile range 0.07 to 0.16), which is also in accordance with the majority of evidence, and that is why we chose this value as the cut-off.
The value of ADC is an aspect that is evaluated in all mpMRI scans, since, as the reviewer comments, it correlates inversely with the GGG. In fact, if there are several lesions in the same patient, lesions with the lowest ADC values are specified in the radiological report. Pitfalls that can cause low ADC values are also considered, as well as low ADC values in non-focal lesions that could reveal sparse tumors. Considering our focus in finding variables to better select PI-RADS 3 lesions that should be biopsied, the role and impact of this value is lower when a PI-RADS 3 lesion has already been diagnosed. According to PI-RADS V.2, these lesions are mildly or moderately hypointense and mildly hyperintense on high b-value DWI; thus, in case of PCa, these lesions usually are low GGG; but, despite this, there also could be csPCa with a high PCa volume.
Reviewer 2 Report
The authors aim to assess the role of concurrent systematic prostate biopsy in addition to targeted sampling for the detection of clinically significant PCa. Before the article would be considered for publication, I recommend some changes, as follows:
- the title does not correctly illustrate the paper - you did not report any role of targeted biopsy in addition to systematic sampling
- the most frequently used abbreviation for clinically significant PCa is csPCa not SigPCa - please change
- SigPCa is only GG2, according to your article - this means that you did not include GG3 or higher?
- please correct "affected cores" with "positive cores"
- MRI was performed with/without endorectal coil?
- what do you understand by mpMRI without enough quality - how did you assess this? Did you use any criteria, such as PIQUAL?
- please detail the biopsy technique: what type of registration was used? Transrectal or transperineal route? How many urologists performed the biopsies? Do you have any information regarding their learning curve? Did you take first the targeted or systematic cores? Was the urologist who performed the systematic sampling blind to the MRI result? Also, please discuss if any of these information could have biased the results of the study
-
Most of the patients had a previous negative biopsy (69.8%) in mpMRI, with a PIRADS score of 4. – please clarify this phrase
- The study is clearly retrospective. It is not correct to say that you included prospectively patients who have already undergone prostate biopsy and the pathological result is available. In order to clarify your methology, I recommend to state as inclusion criteria the indication for prostate biopsy. I assume that you included prospectively all patients who underwent biopsy at the time of the procedure. And then you analyzed retrospectively the data by adding the details regarding pathologic result of the biopsy. Please rewrite this part
Author Response
Response to the Editor and point-by-point responses to the Reviewer’s comments
We sincerely thank the Editor and Reviewers for their constructive comments and suggestions, which we consider to be very helpful in improving the quality of our manuscript. Accordingly, we have made many specific changes in the manuscript, based on these comments, as is explained in detail below in a point-by-point description of the changes introduced. Changes in the revised manuscript are highlighted in red.
We honestly trust that our responses and the modifications provided will help to strengthen the support of the Editor and reviewers, and hope that this revised version of our manuscript fits within the high-quality scientific standards of Diagnostics and, may, therefore, become acceptable for publication in the journal.
Reviewer 2
The authors aim to assess the role of concurrent systematic prostate biopsy in addition to targeted sampling for the detection of clinically significant PCa. Before the article would be considered for publication, I recommend some changes, as follows:
Reviewer 2 comment 1: the title does not correctly illustrate the paper - you did not report any role of targeted biopsy in addition to systematic sampling
Author’s response: We thank the reviewer for pointing this out. Consequently, we have changed the title to what we hope is a more appropriate title.
Reviewer 2 comment 2: the most frequently used abbreviation for clinically significant PCa is csPCa not SigPCa - please change.
Author’s response: We thank the reviewer for this suggestion. Following the reviewer’s suggestion, we have changed this throughout the manuscript.
Reviewer 2 comment 3: SigPCa is only GG2, according to your article - this means that you did not include GG3 or higher?
Author’s response: We sincerely apologize for this mistake and thank you for pointing this out. Effectively, CsPCa was defined as any GGG >2 (Line 78).
Reviewer 2 comment 4: please correct "affected cores" with "positive cores"
Author’s response: We thank the reviewer for pointing this out and we again apologize for that misleading information (Line 78).
Reviewer 2 comment 5: MRI was performed with/without endorectal coil?
Author’s response: We thank the reviewer for this relevant question. The MRI was performed without an endorectal coil. This information has been added in line 83.
Reviewer 2 comment 6: what do you understand by mpMRI without enough quality - how did you assess this? Did you use any criteria, such as PIQUAL?
Author’s response: We thank the reviewer for highlighting this pertinent point. All MRI scans at our center were performed on a single 1.5T MRI scanner with a protocol that fulfilled the minimum technical requirements proposed by PIRADS v2. This MRI protocol meets the criteria of slice thickness, FOV, and in-plane dimension in all sequences. Three acquired b-values and a calculated b value of 1400–2000 sec/mm2 were used for DWI. Also, temporal resolution in the dynamic sequence with contrast was considered.
Despite the fact that, at the time of the study PIQUAL was not yet published, subjective criteria from the most experienced radiologist were used to determine whether an MRI scan was if high enough quality to rule out csPCa. Although acquisition criteria were adequate, some scans were not suitable for evaluation due to patient movements, intestinal peristalsis, or devices, such as a hip prostheses. This information has been added to the manuscript.
Reviewer 2 comment 7: please detail the biopsy technique: what type of registration was used? Transrectal or transperineal route? How many urologists performed the biopsies? Do you have any information regarding their learning curve? Did you take first the targeted or systematic cores? Was the urologist who performed the systematic sampling blind to the MRI result? Also, please discuss if any of these information could have biased the results of the study
Author’s response: We thank the reviewer for these interesting questions. All of this information has been added in the Materials and Methods section, as well as in the Supplemental information, and also a comment about possible limitations in the Discussion section (highlighted in red) Lines109-112, 247-251, 259-302.
Reviewer 2 comment 8: Most of the patients had a previous negative biopsy (69.8%) in mpMRI, with a PIRADS score of 4. – please clarify this phrase
Author’s response: We sincerely apologize for this misunderstanding and thank you for pointing this out. The phrase has been rewritten.
Reviewer 2 comment 9: The study is clearly retrospective. It is not correct to say that you included prospectively patients who have already undergone prostate biopsy and the pathological result is available. In order to clarify your methodology, I recommend to state as inclusion criteria the indication for prostate biopsy. I assume that you included prospectively all patients who underwent biopsy at the time of the procedure. And then you analysed retrospectively the data by adding the details regarding pathologic result of the biopsy. Please rewrite this part
Author’s response: Thank you for your comment. Following the reviewer’s recommendation, this part has been rewritten. Lines 63-75.
Round 2
Reviewer 1 Report
I truly want to see the authors' work published, but the article has many ambiguities within it. Is this about PI-RADS 3 or are the authors covering all PI-RADS cases of any concern ( i.e., 3 - 5?)?
The title remains confusing.
Is this paper about adding standard biopsies to targeted ones? The new title indicates it addresses the issue of possibly both by using "and/or”. And the title is addressing all potentially important PI-RADS lesions from 3 to 5 and then it follows with a second sentence “Should all PI-RADS 3 be biopsied?” Is this paper about PI-RADS 3 cases and what can be done in light of the low positive predictive value to enhance how we select patients?
Given the above, PI-RADS 3 cases present a diagnostic and prognostic challenge. The main objective of our study was to assess the value of adding standard biopsy to targeted biopsy and to evaluate when a lesion biopsy(ies) could be avoided based on lesion location and clinical scenarios when a mp-MRI discloses PI-RADS 3 (i.e., an equivocal or indeterminate lesion) .
I added the comment below to my review of version 2:
I offer this text as a consideration for the authors to ensure clarity in stating the objective of this report.
I am finding too many inconsistencies between citations the authors give and what is in those full text papers and how the authors are relating information.

Author Response
Reviewer 1
Author response: We would like to thank the reviewer for all the comments. We really appreciate the enormous work performed by the reviewer for improving the quality and clarity of our research. We have followed the reviewer’s suggestion and made the changes in the manuscript highlighted in red.
Comments
I truly want to see the authors' work published, but the article has many ambiguities within it. Is this about PI-RADS 3 or are the authors covering all PI-RADS cases of any concern (i.e., 3 - 5?)?
The title remains confusing.
Reviewer 1’ comment1 Is this paper about adding standard biopsies to targeted ones? The new title indicates it addresses the issue of possibly both by using "and/or”. And the title is addressing all potentially important PI-RADS lesions from 3 to 5 and then it follows with a second sentence “Should all PI-RADS 3 be biopsied?” Is this paper about PI-RADS 3 cases and what can be done in light of the low positive predictive value to enhance how we select patients?
Authors response: Thank you for your comment. Following the reviewer’s recommendation, the title has been rewritten in accord with both reviewer comments. The title wants to describe the two main objectives of this research paper. The objective’s paragraph has also been rewritten following the reviewer`s comment.
Reviewer 1’ comment2 objective of our study was to assess the value of adding standard biopsy to targeted biopsy and to evaluate when a lesion biopsy(ies) could be avoided based on lesion location and clinical scenarios when a mp-MRI discloses PI-RADS 3 (i.e., an equivocal or indeterminate lesion) .
I added the comment below to my review of version 2: I offer this text as a consideration for the authors to ensure clarity in stating the objective of this report.
Authors response: Thank you for your comment. Following the reviewer’s recommendation, this paragraphs has been rewritten.
Reviewer 1’ comment3: The Ahmed reference referred to “suspicious” for Likert scale ≥ 3 and not PI-RADS (which had not been published yet). In PI-RADS the value of 3 is ascribed to EQUIVOCAL or INDETERMINATE. I have never seen the word “suspicious” used with PI-RADS 3.
Authors response: We thank the reviewer for pointing this out. We understand the reviewer`s concern and so, we have changed “suspicious” to “positive” as in these three studies either LIKERT or PI-RADS equal or higher than 3 were considered positive to guide and perform target biopsy.
Reviewer 1’ comment4: It would educate the reader if you gave specifics as to these issues regarding lesion location & clinical scenarios, assuming this can be done in a sentence or two.
Authors response: We thank the reviewer for this comment. As along the discussion section these issues are further explained and debated we have considered not to expand in excess the introduction section. We hope it is not an inconvenient for the reviewer.
Reviewer 1’ comment5: I offer this text as a consideration to the authors to ensure clarity in stating the objective of this report.
Authors response: We really appreciate all the reviewer´s suggestion and work. According to that, we have followed the reviewer recommendation and rewritten all the paragraph.
Reviewer 1’ comment6: Clarify if this should be >5 and not ≥ 5 since the authors give a reference (Ahmed) where 6mm or more is considered the significant length. I believe the authors are confusing tumor volume with core length. Their cited references to “5” are 3 papers dealing with tumor volumes of 0.5 ml, not mm. Ahmed: 6 mm or more in any location. Other definitions of clinical significance were also assessed secondarily.
Authors response: We thank the review for this comment. This statement is correct and is our definition of csPCa apart from GGG. We understand this threshold cut-off is in between both Ahmed definitions (1º: more or equal than 6mm and 2ºmore or equal than 4mm) but it is the one used in our cohort. We had also considered number of positive core in this definition but only for standard systematic biopsy.
Reviewer 1’ comment7: Hey, authors, you are talking in this section about Main Variables, not back to the study objectives. And item 2, as written, is not even a complete sentence.
Authors response: We sincerely apologize for this mistake, and thank you for pointing this out. We have moved objectives to the correct place and complete the item 2 sentence also following the reviewer`s previous suggestion.
Reviewer 1’ comment8: Now you are discussing targeted biopsy when just before the findings were about systematic biopsy added to targeted biopsy. I think you create confusion to the reader as to the main objective of your study. Either you need to revise the title further and break up your results into “The value of targeted biopsy” and “The value of systematic biopsy added to targeted biopsy” or only deal with the latter. Otherwise, the message (THLs) you wish to convey to the reader will be lost.
Authors response: We sincerely apologize for this misunderstanding. Now with the title and redaction changes, we hope the messages are clearer.
Reviewer 1’ comment9: I cannot find this discussed in the Falagario paper. Please respond to this critique.
Authors response: We understand the reviewer`s concern. It could be because of a misunderstanding. Falagario et al in their nomogram clearly showed how the probability of finding csPCa in systematic biopsy was lower in more anterior lesions (non-peripheral lesions). We have rewritten the sentences to be in exact accordance with falagario’s paper.
Falagario`s paper stated that “In contrast to TBx, MRI suspicious lesion location and prostate volume were signifcant predictors of csPCa in SBx with p values of 0.039 and 0.02, respectively. The remaining signifcant predictors of csPCa in SBx were similar to TBx” and the prediction probability of csPCa in systematic biopsy of TZ-CZ vs PZ was OR= 0.49 (0.19-0.96) (p= 0.039)
Reviewer 2 Report
Please rephrase the title: "The role of adding standard systematic prostate biopsy to targeted cores to detect ..."
I would rephrase the sentencing in VNav paragraph as follows: " The patient WAS placed" - instead of is placed; "images were imported" - instead of are imported
Author Response
Reviewer 2
We thank the reviewer`s comment and really appreciate his/her support.
Reviewer 2’ comment1: Please rephrase the title: "The role of adding standard systematic prostate biopsy to targeted cores to detect ..."
Authors response: Thank you for your comment. Following the reviewer’s recommendation, the title has been rewritten in accord with both reviewer comments.
Reviewer 2’ comment2: I would rephrase the sentencing in VNav paragraph as follows: " The patient WAS placed" - instead of is placed; "images were imported" - instead of are imported
Authors response: Thank you for your comment. Following the reviewer’s recommendation, these sentences have been changed.